# Ameliorative Effect of Pomegranate Peel Powder on the Growth Indices, Oocysts Shedding, and Intestinal Health of Broilers under an Experimentally Induced Coccidiosis Condition

**DOI:** 10.3390/ani13243790

**Published:** 2023-12-08

**Authors:** Abdul Hafeez, Qambar Piral, Shabana Naz, Mikhlid H. Almutairi, Abdulwahed Fahad Alrefaei, Tugay Ayasan, Rifat Ullah Khan, Caterina Losacco

**Affiliations:** 1Department of Poultry Science, Faculty of Animal Husbandry and Veterinary Sciences, The University of Agriculture, Peshawar 25000, Pakistan; hafeez@aup.edu.pk (A.H.);; 2Department of Zoology, Government College University, Faisalabad 38000, Pakistan; drshabananaz@gcuf.edu.pk; 3Department of Zoology, College of Science, King Saud University, P.O. Box 2455, Riyadh 11451, Saudi Arabia; malmutari@ksu.edu.sa (M.H.A.); afrefaei@ksu.edu.sa (A.F.A.); 4Faculty of Applied Sciences, Osmaniye Korkut Ata University Kadirli, Osmaniye 80000, Turkey; tayasan@gmail.com; 5Faculty of Animal Husbandry and Veterinary Sciences, College of Veterinary Sciences, The University of Agriculture, Peshawar 25000, Pakistan; 6Department of Precision and Regenerative Medicine and Jonian Area, Section of Veterinary Science and Animal Production, University of Bari Aldo Moro, 70010 Bari, Italy; caterina.losacco@uniba.it

**Keywords:** broilers, coccidiosis, lesion score, cecum histology, pomegranate peel powder

## Abstract

**Simple Summary:**

This research investigated the feasibility of using discarded pomegranate peel powder as a natural supplement to enhance growth and gut health in broilers, offering a natural solution against coccidiosis. The inclusion of 3 and 6 g/kg of pomegranate peel powder demonstrated significant efficacy in alleviating the hindered growth rate induced by coccidial oocysts. These doses also led to a notable reduction in pathological lesions within the cecum and a decrease in oocyst numbers. Additionally, they restored the cecal morphological features in the broilers. These results emphasize the beneficial potential of incorporating 3 and 6 g/kg of pomegranate peel powder as a dietary supplement, potentially enhancing the growth and cecum health of broilers in response to coccidial challenge. Thus, pomegranate peel provides a natural, environmentally friendly, and cost-effective solution for poultry production under a coccidial challenge.

**Abstract:**

Coccidiosis stands as one of the most prevalent enteric parasitic diseases in broilers. While antibiotics have traditionally been used for the control of coccidiosis, concerns related to drug residues and the emergence of resistance in chickens have prompted consumer apprehensions. In this study, 600 Ross 308 broiler chicks were randomly divided into five groups: a control group without specific treatments (NC), broilers deliberately exposed to *Eimeria tenella* (positive control), broilers challenged with *E. tenella* but dosed with antibiotics (AT), and two groups experimentally exposed to *E. tenella* while simultaneously receiving pomegranate peel powder (PPE) at dosages of 3 g/kg (3PPP) and 6 g/kg (6PPP). The results revealed that all *Eimeria*-treated birds exhibited significantly worse growth performance compared to the NC. Notably, a marked improvement was observed in birds infected with *E. tenella* when supplemented with 6PPP. Both 3PPP and 6PPP supplementation significantly reduced lesion scores, mortality, and oocysts per gram (OPG). Furthermore, histological examination of the cecum indicated that the villus dimensions were restored by PPP supplementation in infected birds. In conclusion, *Eimeria*-infected birds supplemented with 6PPP experienced an enhanced growth rate, lowered lesion scores, alleviated oocyst shedding, and improved intestinal histological dimensions.

## 1. Introduction

Coccidiosis, caused by a number of *Eimeria* species, is a significant parasitic disease known for pathological lesions in the intestines [1]. This disease holds economic importance as one of the most prevalent and costly parasitic afflictions affecting the broiler industry. Typically, *Eimeria* proliferates, causing severe alterations in the epithelial tissue of various intestinal segments through multiplication [2]. The global prevalence of coccidiosis in countries with substantial poultry production leads to substantial economic losses, encompassing reduced weight, diminished feed efficiency, expenses related to anti-coccidial therapy, and increased mortality and morbidity [2,3]. Additionally, *Eimeria* infections are associated with the disruption of digestion and absorption of vital nutrients, leading to reduced growth performance and immunocompromised states [4]. 

In the current scenario, the primary strategy to combat coccidiosis on poultry farms includes employing antibiotics, vaccination, natural compounds [2,5,6], or a combination of these approaches. Although these strategies have demonstrated high effectiveness in managing economically important parasitic infections, apprehensions have been raised regarding the emergence of antibiotic resistance in meat and eggs [7,8]. Consequently, researchers have focused on finding health-conscious solutions as an alternative to antibiotics [9]. As a matter of fact, in the developed parts of the world, the application of antibiotics for the prevention of infectious diseases has been significantly reduced for more than a decade and a total removal is expected in the coming years [10].

Numerous research reports have delved into the beneficial impact of herbal-based compounds on broiler performance, intestinal physiology, and the reduction in mortality. In recent times, with the goal of controlling coccidiosis, phytogenic compounds have emerged as a promising alternative to antibiotics, not only demonstrating efficacy in reversing *Eimeria* infection but also showing little negative effects on bird health [11]. Recent research findings have concluded the anti-*Eimeria* characteristics of phytogenic compounds, such as powders and extracts [1,5,6]. While the application of phytogenics represents an important option for reversing the *Eimeria* challenge without the use of antibiotics in broiler production [12], their adoption by poultry farmers could be enhanced if these products are derived from agricultural wastes, providing an even more potent alternative to antibiotics [5].

Pomegranate (*Punica granatum*) peel is rich in phenolic compounds, endowing it with higher antioxidant properties compared to other parts of the fruit. Additionally, it is known for its attributed anti-fungal, anti-parasitic, and anti-cancer properties [13]. The fruit of pomegranate itself contains an array of valuable components including amino acids, minerals, various acidic compounds, antioxidants (punicalagin, anthocyanins, and ellagic acid), and anti-cancer agents [14]. Studies in the past have also demonstrated the anti-parasitic potential of pomegranate peel, showing that it acts on the different stages of growth and development in parasites. Notable examples include its impact on the adult parasites of *Trichomonas tenax* [15], *Schistosoma mansoni* [16], *Giardia* [17], and *Cryptosporidium parvum* [18], as well as certain nematodes and *strongyles* [19].

Based on prior research, there is evidence to suggest that pomegranate peel exhibits notable anti-parasitic properties against *Eimeria*, a phenomenon explored in this particular study. Therefore, our research focused on assessing the impact of pomegranate peel powder on *Eimeria* by using various parameters including decreased oocyst excretion, diminished intestinal lesions, microscopic alterations in villi, and the influence of treatment on the growth metrics of broiler chickens.

## 2. Materials and Methods 

### 2.1. Preparation of Pomegranate Peel Powder (PPP) 

Fresh pomegranates were obtained from the local grocery shop. The edible portion of the fruit was extracted, and the peels were air-dried under a shadow and then milled to powdered form.

### 2.2. Preparation of Eimeria Oocysts

Briefly, from the feces of infected broilers, *Eimeria tenella* oocysts were extracted, mixed with a K_2_Cr_2_O_3_ solution (2.5%), and finally incubated at 25 °C. The progress of sporulation was assessed daily [5]. Once sporulated, the oocysts underwent several centrifugation and washing steps until the supernatant appeared. This layer was carefully removed and stored for subsequent use at 4 °C. For the inoculation process, the suspended oocysts were rinsed twice with distilled water, followed by adjustment to a concentration of at least 1 × 10^4^ oocysts/mL with the help of the McMaster method [10].

### 2.3. Bird Husbandry and Experimental Design

In the current trial, a total of 600 broiler chicks (Ross 308; initial weight 43 ± 0.25 g) were carefully prepared and subsequently randomly allocated to 20 pens using a completely randomized design for a duration of 35 days. Each pen housed 15 birds. Uniform nutritional and management conditions were maintained for all chickens up to the age of 21 days. The birds were exposed to a 20 h light/day cycle and had ad libitum access to both feed (Table 1) and fresh drinking water.

The experimental groups were defined as follows:The control group (NC) broilers that did not receive any specific treatments.*E. tenella* challenged broilers (positive control).*E. tenella* challenged (AT) broilers that also received an antibiotic, Amprolium (1 g/kg).*E. tenella* challenged broilers concurrently fed with pomegranate peel powder at a rate of 3 g/kg (3PPP).*E. tenella* challenged broilers that received a PPP at a dosage of 6 g/kg (6 PPP).

To mitigate the risk of cross-contamination between infected and uninfected pens, distinct plastic boots were assigned to each group. One set of boots exclusively served the pens exposed to the *Eimeria* challenge, while the other set was designated for the uninfected pens. Additionally, trays containing CaO were strategically placed at the entrance of the uninfected pens to decrease the likelihood of potential contamination, acting as an additional barrier.

### 2.4. Coccidial Infection

On day 22 of the trial, except for the NC group, all other groups were challenged with a dose of 5 × 10^4^ sporulated *Eimeria tenella* oocysts/mL/bird orally. Strict measurements were imposed to decrease any chance of contact between the exposed and control broilers, despite their close proximity in housing. Mortality rates were regularly recorded in the datasheet to monitor the possible deaths following the induced *Eimeria* challenge. To maintain balance, the NC group was administered an equivalent volume of normal saline orally.

### 2.5. Performance Traits

On a weekly basis, the feed intake (FI), weight gain (WG), and feed conversion ratio (FCR) were assessed for the entire period (35 days) of the experiment. 

### 2.6. Lesion Score

The severity of lesions in the cecum was assessed using the grading system established by Khorrami et al. [2]. The evaluation was conducted on a scale of 0 to 4 as follows: 0 (no lesions), 1 (mild), 2 (moderate), 3 (severe), and 4 (very severe).

### 2.7. Oocyst Counting

Oocyst counting was performed by collecting fecal samples from each cage on days 5, 7, and 9 post-infection (dpi), following the methodology suggested by Khan et al. [1]. The determination of oocysts per gram (OPG) was conducted using the McMaster chamber technique, with counting performed by using a compound microscope (Nikon, Japan) at 10× magnification.

### 2.8. Pathological Investigation

To assess pathological lesions at the histological level, four cecal samples per cage were aseptically obtained following the protocol outlined by Khan et al. [1]. The procedure involved extracting a small portion from the cecum, sectioning it up to 1 cm^2^, and placing it in a buffered formalin solution (10%). The tissue samples were dehydrated with an alcohol solution. Ultimately, tissue blocks were prepared, and the tissues were sliced using a microtome. The tissue samples underwent H&E staining. The slides were evaluated under a microscope for recording villi dimensions. 

### 2.9. Statistical Analysis

The data were analyzed through a one-way analysis of variance (ANOVA). To evaluate mean significance, Tukey’s test was employed with a predetermined significance level of *p* < 0.05. Statistical analysis was conducted using STATISTIC-2010, version 8.1, software, with each pen serving as an experimental unit. 

## 3. Results

Table 2 illustrates that feed intake was not significantly (*p* > 0.05) affected in weeks 1–3, or during the starter phase. However, in week 4, the feed intake in the NC group was significantly (*p* < 0.05) higher than in the AT group, followed by the PPP3 and PPP6 groups, which were, in turn, significantly higher (*p* < 0.05) than the PC group. Similarly, in week 5, the feed intake of the NC group was significantly (*p* < 0.05) higher than in the AT group, followed by the PPP6, PPP3, and PC groups. At the end of the finisher phase, the feed intake remained higher (*p* < 0.05) in the NC group compared to the AT group, followed by the PPP6, PPP3, and PC groups. Overall, the feed intake of the NC group was significantly (*p* < 0.05) higher than the AT group, followed by the PPP6 and PPP3 groups, which were again significantly (*p* < 0.05) higher than the PC group. 

Table 3 shows the impact of pomegranate peel powder on the weight gain of broilers challenged with *E. tenella*. At week 1 the weight gain recorded was not affected (*p* > 0.05) by the inclusion of PPP in the diets. At week 2, the weight gain was higher (*p* < 0.05) in the broilers of the PPP6 and PPP3 groups, followed by the AT, PC, and NC groups. In week 3, it was observed that the WG of the PPP6 group was significantly (*p* < 0.05) higher than the PPP3 and NC groups, followed by the PC and AT groups. Furthermore, the PPP3 group exhibited a significantly (*p* < 0.005) higher weight gain than both the PC and AT groups. During the starter phase, the weight gain was significantly (*p* < 0.05) higher in the PPP6 group compared to the PPP3 group, followed by the NC, AT, and PC groups. In week 4, the weight gain in the NC group was significantly (*p* < 0.05) higher than in the AT and PPP3 groups, followed by the PPP3 and PC groups. The week 5 weight gain showed that the NC group was significantly higher than the AT group, followed by the PPP6, PPP3, and PC groups. In the finishing phase, the weight gain results indicated that the NC group exhibited significantly (*p* < 0.05) higher weight gain compared to the AT group, followed by the PPP6, PPP3, and PC groups. Overall, the weight gain in the NC group was significantly (*p* < 0.05) greater than the AT group, followed by the PPP6, PPP3, and PC groups.

Table 4 shows that the FCR in the 1st and 2nd weeks was significantly similar (*p* > 0.05). In the third week, the FCR in the PPP6 group was significantly (*p* < 0.05) improved compared to the PPP3 group, followed by the NC, PC, and AT groups. During the starter phase, the feed conversion ratio (FCR) was significantly (*p* < 0.05) lower in the PPP6 group compared to the PPP3 group, followed by the NC, AT, and PC groups. In the fourth week, the FCR in the PPP6 group was significantly (*p* < 0.05) lower than in the NC group, which was further reduced than in the PPP3 and AT groups, followed by the PC group. By the fifth week, the FCR was significantly (*p* < 0.05) lower in the NC group than the AT group, with no difference in the PPP3 and PPP6 groups, which were significantly (*p* < 0.05) less improved than the PC group. In the finisher phase, the FCR was significantly (*p* < 0.05) lower in the NC and PPP6 groups compared to the AT and PPP3 groups, followed by the PC groups. Over the entire period, the FCR was significantly (*p* < 0.05) lower in the PPP6 and NC groups compared to the AT and PPP3 groups, followed by the PC group.

Table 5 illustrates the impact of PPP supplementation on lesion scores and mortality in broilers subjected to an *E. tenella* challenge. The NC group, which remained uninfected, exhibited a normal cecal epithelium. In contrast, the PC group showed significantly (*p* < 0.05) higher cecal lesions compared to the PPP3 group, followed by the PPP6 group and the AT group. Lesions were absent in the NC group, and the AT-treated group displayed the lowest lesions. Mortality rates were highest in the PC group, followed by the PPP3 and PPP6 groups. No mortality occurred in the NC and AT groups.

Table 6 presents the effects of pomegranate peel powder (PPP) supplementation on the oocyst count per gram (OPG) of feces in broilers subjected to an *E. tenella* challenge. The OPG, in comparison to the NC and AT groups, was significantly (*p* < 0.01) higher in the PC group on days 5, 7, and 9 post-infection (DPI). The OPG on day 7 DPI remained unchanged (*p* > 0.05) between the PC group and PPP supplemental groups. However, in the second and third intervals (10 and 14 days) of the challenge, the OPG count significantly (*p* < 0.05) decreased in the PPP supplemental groups compared to the PC group.

Table 7 illustrates the effect of supplementing broilers challenged with coccidiosis with PPP on the villus dimensions of the caecum. Significantly decreased villus height, width, and VH:CD were observed in the PC group compared to the NC and AT groups (*p* < 0.05). Interestingly, the PPP3 and PPP6 groups exhibited significantly higher villus height, width, and VH:CD ratios compared to the PC group (*p* < 0.05). Similarly, the crypt depth significantly decreased in the PPP3 and PPP6 groups compared to the PC group (*p* < 0.05). Examination of the cecal epithelium (Figure 1) in the PC group revealed prevalent signs of scattered *Eimeria* oocysts, desquamation of the epithelium, aggregation of inflammatory cells, and hemorrhages. In contrast, both pomegranate peel powder-supplemented groups (at 3 g/kg and 6 g/kg) showed minimal pathological lesions and mild sloughing in the epithelium of the cecum of infected broilers. The broilers that were AT-treated after infection showed only minor damage to the cecum epithelium.

## 4. Discussion

In the current study, the effects of pomegranate peel powder (PPP) on the growth indices, oocysts excretion, and cecal histomorphology of broiler chickens subjected to induced *Eimeria* challenge were evaluated. The results showed that PPP supplementation reduced lesion scores following *Eimeria* infection, led to a decrease in *Eimeria* oocyst numbers, and restored the cecal histological morphology of infected broilers. Notably, the impact of PPP6 was superior to PPP3 in terms of growth performance, with broilers challenged with *Eimeria* exhibiting improved growth performance when their diet was enriched with PPP6. Khorrami et al. [2] also reported improved growth performance and alleviation of coccidiosis symptoms in broiler chickens challenged with mixed *Eimeria* species in response to 200 and 400 ppm pomegranate peel extract (PPE). Several studies have also explored the impact influence of PPP on growth indices in chickens, with Akuru et al. [20] finding improved weight gain and feed efficiency in birds supplemented with 2 g/kg and 4 g/kg of PPP. Hamady et al. [21] conducted a study involving the addition of 0.1% pomegranate peel extract powder in broilers’ diet for a period of six weeks. The results demonstrated that continuous supplementation led to an increase in weight gain. Another study by Rezvani and Rahimi [22] investigated the effects of PPEx on various parameters including the weight gain, nutrient digestibility, immune response, and microbial population of broiler chickens. Their findings indicated that PPEx enhanced the daily weight gain and feed intake throughout the growing period. However, it did not have a significant impact on the FCR. Additionally, the extract of pomegranate peels was found to improve the digestibility of nutrients, promote the growth of beneficial microbial flora like Lactobacillus, and elevate the immune response [22]. This enhancement is likely attributed to the growth-promoting properties of pomegranate peel, which are linked to its antimicrobial and antioxidant attributes. The presence of proanthocyanidins in pomegranate peel facilitates improvements in pancreatic and small intestinal digestive enzyme functions. Additionally, it helps counteract the detrimental effects of reactive oxygen species on intestinal enterocytes, ultimately resulting in improved nutrient digestion [23,24]. 

In this investigation, both supplements exhibited comparable efficacy in relation to OPG, lesion scores, and villus histology. Furthermore, the evaluation of PPP3 and PPP6 doses indicated that, although they demonstrated some anti-*Eimeria* effects, these effects were not parallel to the AT treatment concerning the mitigation of typical coccidiosis symptoms and the enhancement of growth performance. 

Researchers posit that integrating phenolic compounds into the food chain can serve as an effective strategy for controlling parasitic diseases. It represents an affordable and straightforward means to enhance the health of poultry in their defense against parasitic infections. Specifically, pomegranate peel stands out as a valuable reservoir of bioactive compounds [25]. In a study conducted by Elfalleh et al. [26], various components of the pomegranate plant were compared for their levels of flavonoids, total phenolics, anthocyanins, and hydrolysable tannins. The study revealed that the pomegranate peel exhibited the highest concentration of these compounds. Various studies have also demonstrated the in vivo and in vitro antimicrobial and anti-parasitic properties of pomegranate peel [26]. In addition, its anti-fungal effect against *Aspergillus niger* and its anti-bacterial effect against *Staphylococcus aureus* have been reported [27]. Furthermore, its anti-parasitic effect on the adult worms of *S. mansoni* and *Trichomonas tonax* has been confirmed [15,16]. Al-Megrin [28] also documented the anti-cestodal activity of pomegranate peel extract against *Hymenolepis nana*, which is an infection caused by a dwarf tapeworm [28]. Yet another study provided evidence for the efficacy of pomegranate peel extract in both the prevention and treatment of *Giardia lamblia* infection [17]. 

Pomegranate exhibits anti-coccidial activity similar to Amprolium, as evidenced by a significant reduction in the excretion of *E. tenella* oocysts in the feces of infected birds. This reduction indicates that pomegranate hampers the development of parasites within the host before the formation and eventual release of the relatively inert oocysts [9]. The antioxidative properties of pomegranate are primarily attributed to its phenolic compounds, including gallotannins, anthocyanins, gallagyl esters, ellagitannins, hydroxycinnamic acids, hydroxybenzoic acids, and dihydroflavonols [29,30]. These compounds in pomegranate have been reported to exhibit anti-cestodal, anti-nematodal [31], anti-protozoan [32,33], and anti-bacterial effects [34,35]. In our study, the administration of a PPP-fortified diet resulted in an enhanced histological picture of the challenged broilers, which can be linked to the anti-protozoal effects of PPP, coupled with its potential anti-inflammatory and antioxidant activities. These attributes likely played a significant role in protecting the host tissue from damage caused by *Eimeria* oocysts [35]. The effectiveness of pomegranate extracts has primarily been attributed to their major chemical component, anthocyanins. This compound has been reported to exhibit both anti-coccidial and anti-protozoal activities [18]. As natural products, pomegranates show considerable promise as potential sources for novel anti-coccidial agents to mitigate *Eimeria* infection. 

## 5. Conclusions

The study concluded that feed supplemented with pomegranate peel powder at 6 g/kg significantly improved growth performance, reduced *Eimeria* oocyst excretion in feces, and reversed histopathological lesions in broilers exposed to *E. tenella* infection.

## Figures and Tables

**Figure 1 animals-13-03790-f001:**
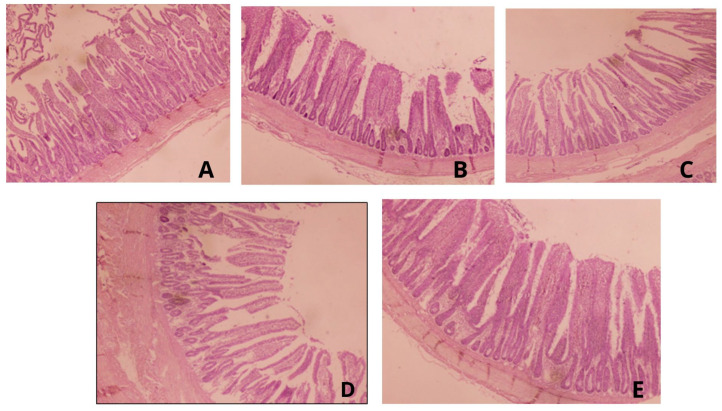
(**A**) Figure of the cecal villus from the NC broilers showing nearly normal villi. (**B**) Figure of the cecal villus from the PC broilers displaying erosion as well as desquamation of the crypt, along with inflammatory cells in the lamina propria. (**C**) Figure of cecal villus-challenged + AT treated broilers showing villi similar to NC broilers. (**D**,**E**) Figures of PPP-treated broilers showing a mild erosion of the epithelium and less desquamation of epithelium in *Eimeria*-infected broilers (HE × 200).

**Table 1 animals-13-03790-t001:** Composition and chemical analysis of the basal diets.

Ingredients	Starter Phase (1–21 Days)	Finisher Phase (22–35 Days)
Yellow corn	53.21	60.75
Soybean meal (48% CP)	37.92	25.00
Corn gluten meal (34% CP)	2.00	7.10
Corn oil	2.20	2.80
Dicalcium phosphate	2.30	2.05
Limestone	0.83	0.68
NaCl	0.45	0.50
Vitamin and mineral premix (standard level)	0.50	0.50
DL-Methionine	0.20	0.10
L-Lysine HCl	0.22	0.37
L-Threonine	0.11	0.10
Choline chloride	0.05	0.05
Chemical composition		
ME, kcal/kg	3000	3150
Crude protein, %	22.5	21.30
Methionine, %	0.55	0.44
Lysine, %	1.42	1.23
Sulfur amino acids, %	0.96	0.80
Threonine, %	0.95	0.85
Calcium, %	1.05	0.90
Available phosphorus, %	0.50	0.45

**Table 2 animals-13-03790-t002:** The impact of supplementing broiler diets challenged with *E. tenella* with pomegranate peel powder on feed intake (g).

	NC	PC	AT	PPP3	PPP6	SEM	*p*-Value
Week 1	123.6	123.5	124.5	125.4	125.4	0.85	0.935
Week 2	341.5	343.8	338.4	343.7	343.4	0.78	0.188
Week 3	592.8	589.9	591.2	593.8	592.3	0.716	0.446
Starter phase	1056.7	1055.4	1053.4	1061.9	1059.1	1.612	0.491
Week 4	790.6 ^a^	652.1 ^d^	781.9 ^b^	711.6 ^c^	710.6 ^c^	13.68	<0.001
Week 5	963.4 ^a^	799.4 ^e^	932.6 ^b^	884.4 ^d^	901.8 ^c^	14.81	<0.001
Finisher phase	1753.1 ^a^	1451.5 ^e^	1713.4 ^b^	1595.4 ^d^	1611.9 ^c^	28.18	<0.001
Overall mean	2809 ^a^	2506 ^d^	2766 ^b^	2656 ^c^	2671.5 ^c^	28.12	<0.001

Means with dissimilar superscripts in a row differ significantly (*p* < 0.01).

**Table 3 animals-13-03790-t003:** Impact of pomegranate peel powder supplementation on weight gain (g) in broilers challenged with *E. tenella*.

Group	NC	PC	AT	PPP3	PPP6	SEM	*p*-Value
Week 1	104.4	104	106.9	106.4	105.7	0.535	0.628
Week 2	265.5 ^b^	266 ^b^	266.6 ^b^	271.4 ^ab^	273.5 ^a^	0.975	0.006
Week 3	376.7 ^bc^	373 ^c^	372.5 ^c^	380.6 ^b^	387.4 ^a^	1.59	<0.001
Starter phase	745.9 ^c^	743 ^c^	744.5 ^c^	757.5 ^b^	765.3 ^a^	2.39	<0.001
Week 4	474.6 ^a^	291 ^d^	443.3 ^b^	407.5 ^c^	438.7 ^b^	16.96	<0.001
Week 5	543.4 ^a^	343 ^e^	517.2 ^b^	481.1 ^d^	486.8 ^c^	18.51	<0.001
Finisher phase	1017.1 ^a^	634 ^e^	960.8 ^b^	888.1 ^d^	924.9 ^c^	35.37	<0.001
Overall mean	1762.3 ^a^	1377.4 ^e^	1704.5 ^b^	1645.9 ^d^	1690.8 ^c^	35.97	<0.001

Means with dissimilar superscripts in a row differ significantly (*p* < 0.01).

**Table 4 animals-13-03790-t004:** The influence of supplementing broilers challenged with *E. tenella* with pomegranate peel powder on the feed conversion ratio.

Group	NC	PC	AT	PPP3	PPP6	SEM	*p*-Value
Week 1	1.18	1.18	1.16	1.18	1.19	.008	0.91
Week 2	1.29	1.29	1.27	1.27	1.25	0.005	0.0690
Week 3	1.58 ^a^	1.58 ^a^	1.59 ^a^	1.56 ^a^	1.53 ^b^	0.006	0.0100
Starter	1.42 ^a^	1.42 ^a^	1.42 ^a^	1.40 ^ab^	1.38 ^b^	0.004	0.0200
Week 4	1.67 ^c^	2.24 ^a^	1.76 ^b^	1.75 ^b^	1.62 ^d^	0.059	<0.01
Week 5	1.77 ^d^	2.33 ^a^	1.80 ^c^	1.84 ^b^	1.85 ^b^	0.055	<0.01
Finisher	1.72 ^c^	2.29 ^a^	1.78 ^b^	1.80 ^b^	1.74 ^c^	0.057	<0.01
Overall	1.59 ^c^	1.82 ^a^	1.62 ^b^	1.62 ^b^	1.58 ^c^	0.024	<0.01

Means with dissimilar superscripts in a row differ significantly (*p* < 0.01).

**Table 5 animals-13-03790-t005:** The influence of *E. tenella* on the lesion scores and mortality of broilers on a diet supplemented with pomegranate peel powder.

Groups	Lesion Scoring	Mortality (%)
NC	0.0 ^e^ ± 0.00	0.00 ^d^ ± 0.00
PC	2.6 ^a^ ± 0.18	22.5 ^a^ ± 0.23
AT	0.3 ^c^ ± 0.01	0.0 ^d^ ± 0.00
PPP3	1.6 ^ab^ ± 0.08	17.5 ^b^ ± 0.16
PPP6	1.0 ^bc^ ± 0.00	10.0 ^c^ ± 0.24
*p*-value	<0.001	0.02 ± 0.00

Means with dissimilar superscripts in a column differ significantly (*p* < 0.01).

**Table 6 animals-13-03790-t006:** The impact of supplementing broilers challenged with *E. tenella* with pomegranate peel powder on the oocyst count per gram of their feces.

Groups	5 DPI	7 DPI	9 DPI
NC	00 ^c^ ± 00	00 ^c^ ± 0.00	00 ^b^ ± 0.00
PC	248.3 ^a^ ± 18.12	557.1 ^a^ ± 16.32	291.6 ^a^ ± 12.68
AT	101.2 ^b^ ± 2.34	181.0 ^b^ ± 12.42	105.6 ^b^ ± 15.84
PPP3	148.1 ^ab^ ± 10.32	224.6 ^b^ ± 1.36	136.3 ^b^ ± 1.38
PPP6	145.2 ^ab^ ± 10.44	212.3 ^b^ ± 13.63	135.3 ^b^ ± 4.55
*p*-value	0.0022	0.0023	0.0112

Means with dissimilar superscripts in a column differ significantly (*p* < 0.01).

**Table 7 animals-13-03790-t007:** The impact of supplementing broilers challenged with *E. tenella* with pomegranate peel powder on the villus dimensions of the caecum.

Groups	Villus Height (μm)	Crypt Depth (μm)	VH:CD	Villus Width (μm)
NC	842.2 ^a^ ± 12.02	199.0 ^c^ ± 3.56	5.1 ^a^ ± 0.21	161.3 ^a^ ± 9.67
PC	827.6 ^c^ ± 11.01	216.3 ^a^ ± 4.36	1.7 ^c^ ± 0.19	123.7 ^c^ ± 8.12
AT	836.7 ^a^ ± 16.01	201.5 ^c^ ± 5.78	4.9 ^a^ ± 0.25	156.6 ^ab^ ± 10.91
PPP3	832.2 ^ab^ ± 14.01	204.9 ^b^ ± 6.98	3.1 ^b^ ± 0.22	148.6 ^b^ ± 7.81
PPP6	831.3 ^b^ ± 13.01	203.5 ^b^ ± 3.77	3.1 ^b^ ± 0.24	145.7 ^b^ ± 6.31
*p*-value	0.0003	0.0234	0.0404	0.0025

Means in the same column but with dissimilar superscripts are significantly different at *p* < 0.05.

## Data Availability

Data are contained within the article.

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
