# Peer review of "Ameliorative Effect of Pomegranate Peel Powder on the Growth Indices, Oocysts Shedding, and Intestinal Health of Broilers under an Experimentally Induced Coccidiosis Condition"

_animals, 2023, doi:10.3390/ani13243790_

Round 1

Reviewer 1 Report

Comments and Suggestions for Authors

Ameliorative Effect of Pomegranate Peel Powder on Growth Indices, Oocysts Shedding, and Intestinal 2 Health of Broilers Under Experimentally Induced Coccidiosis Condition

COMMENTS TO ADDRESS

GENERAL COMMENTS

The authors mentioned Japanese quails in the summary which does not align with the present study. Please explain the linkage with the present study.

Additionally, the methodology lacks experimental design and this must be included to make the work acceptable for publication.

Furthermore, the statistical analysis section could be improved to enhance its readability.

SUMMARY

The study was about broiler chickens (Ross 308), why was the mention of Japanese quails in the summary section?

Check these two sentences:

“Additionally, it played a crucial role in restoring the cecal morphology of Japanese quails”.

……….potentially contributing to the overall well-being and performance of quails………..

INTRODUCTION

L10-11: Delete “caused by various species of Eimeria”

L20: Revise “chemicals drugs” to chemical drugs.

L26: Recast this phrase “…..been severely restricted for the last more than one decade”

L57-58: Please provide the source of the pomegranate peels.

L68: 2.3 Birds husbandry and experimental design. Please provide the average initial weight of the birds.

L75-76: Delete “from” and place “Huvepharma Inc., Peachtree 75 City, GA, USA” in parenthesis.

L93-94: Revise “Throughout the entire growth period, we assessed key growth indices including feed intake (FI), 93 weight gain (WG), and feed conversion ratio (FCR) on a weekly basis” to remove “we”. Suggestion: “Throughout the entire growth period, the key growth indices assessed included feed intake (FI), weight gain (WG), and feed conversion ratio (FCR) on a weekly basis.

L131: Table 2

I have a problem with the way the authors presented the results of the feed intake. There should be a footnote to explain the starter phase and finisher phase. What does the overall mean indicate and are the authors indicating that the birds consume on average 2.6 kg over a 6-week period? If the low values are a result of the E. tenella, then the NC birds were expected to have consumed more feed.

L131: Revise “At week1 the weight gain was significantly (P>0.05) not affected” to At week1 the weight gain recorded was not affected (P>0.05) by the inclusion of PPP in the diets.

L139: Replace “recorded” with observed.

L149: Table 3. Please check the table formatting. Again, at what week was the experiment terminated? The weight gain values recorded are too low compared to the breeder guide for Ross 308. What might be causing these very low weight gains?

L155: Delete the week after the 1st and add “s” to the week after the 2nd. Again, delete “was significantly (P>0.05) not affected” and replace with “were similar (P>0.05)”. Further, revise “At 3rd week” to “In the 3rd week……

L156: If FCR is lesser, then it means it improved so revise the statement appropriately.

L174: Revise “compare” to compared.

Table 5: Check the p-vale for the mortality again. It can’t be “0.02± 0.01

L196: Start “Effect” with a lowercase.

L221: Please find an alternative phrase for “In the present study”.

L254 and 255: “Aspergillus niger” and “Staphylococcus aureus” should be in italics because they are scientific names.

L256: First mentioning “S. mansoni” please write in full and also in italics including “Trichomonas tonax”.

L258: Check previous comments on the scientific naming for “Hymenolepis nana”.

L260: Please italicize “Giardia lamblia”.

L269: Please italicize “E. tenella”.

Comments on the Quality of English Language

The English language is generally good, and the only issue is with the excessive use of "in this study" in the results section.

Author Response

Dear academic reviewer

Thank you very much for reviewing our paper. The comments are highly constructive and beneficial for our paper. The paper has been improved in view of these comments. We have revised the paper according to the reviewer comments. We hope that the revised version will be acceptable to you. The point by point response is as follow:

Reviewer 1

GENERAL COMMENTS

The authors mentioned Japanese quails in the summary which does not align with the present study. Please explain the linkage with the present study. Additionally, the methodology lacks experimental design and this must be included to make the work acceptable for publication. Furthermore, the statistical analysis section could be improved to enhance its readability.

Response: Thank you very much for your comments. We have revised the paper according to the comments.

SUMMARY

The study was about broiler chickens (Ross 308), why was the mention of Japanese quails in the summary section?

Response: corrected…………thanks

Check these two sentences:

“Additionally, it played a crucial role in restoring the cecal morphology of Japanese quails”.

……….potentially contributing to the overall well-being and performance of quails………..

Response: corrected……….thanks

INTRODUCTION

L10-11: Delete “caused by various species of Eimeria”

Response: deleted……..thanks

L20: Revise “chemicals drugs” to chemical drugs.

Response: corrected……..thanks

L26: Recast this phrase “…..been severely restricted for the last more than one decade”

Response: Rephrased………thanks

L57-58: Please provide the source of the pomegranate peels.

Response: added………thanks

L68: 2.3 Birds husbandry and experimental design. Please provide the average initial weight of the birds.

Response: added………thanks

L75-76: Delete “from” and place “Huvepharma Inc., Peachtree 75 City, GA, USA” in parenthesis.

Response: deleted……….thanks

L93-94: Revise “Throughout the entire growth period, we assessed key growth indices including feed intake (FI), 93 weight gain (WG), and feed conversion ratio (FCR) on a weekly basis” to remove “we”. Suggestion: “Throughout the entire growth period, the key growth indices assessed included feed intake (FI), weight gain (WG), and feed conversion ratio (FCR) on a weekly basis.

Response: revised as suggested……….thanks

L131: Table 2

I have a problem with the way the authors presented the results of the feed intake. There should be a footnote to explain the starter phase and finisher phase. What does the overall mean indicate and are the authors indicating that the birds consume on average 2.6 kg over a 6-week period? If the low values are a result of the E. tenella, then the NC birds were expected to have consumed more feed.

Response: Overall mean indicate the feed intake over the entire period of study (35 days). Obviously, the NC birds have consumed more feed 2809 compared 2506 in PC group. Thanks

L131: Revise “At week1 the weight gain was significantly (P>0.05) not affected” to At week1 the weight gain recorded was not affected (P>0.05) by the inclusion of PPP in the diets

Response: revised……..thanks .

L139: Replace “recorded” with observed.

Response: Replaced……..thanks

L149: Table 3. Please check the table formatting. Again, at what week was the experiment terminated? The weight gain values recorded are too low compared to the breeder guide for Ross 308. What might be causing these very low weight gains?

Response: we have now added the total experimental period of 35 days. We have revised these mistakes…..thanks

L155: Delete the week after the 1st and add “s” to the week after the 2nd. Again, delete “was significantly (P>0.05) not affected” and replace with “were similar (P>0.05)”. Further, revise “At 3rd week” to “In the 3rd week……

Response: revised………thanks

L156: If FCR is lesser, then it means it improved so revise the statement appropriately.

Response: revised……..thanks

L174: Revise “compare” to compared.

Response: revised…..thanks

Table 5: Check the p-vale for the mortality again. It can’t be “0.02± 0.01”

Response: corrected…….thanks

L196: Start “Effect” with a lowercase.

Response: corrected……thanks

L221: Please find an alternative phrase for “In the present study”.

Response: corrected………thanks

L254 and 255: “Aspergillus niger” and “Staphylococcus aureus” should be in italics because they are scientific names.

Response: corrected………thanks

L256: First mentioning “S. mansoni” please write in full and also in italics including “Trichomonas tonax”.

Response: corrected………thanks

L258: Check previous comments on the scientific naming for “Hymenolepis nana”.

Response: corrected………thanks

L260: Please italicize “Giardia lamblia”.

Response: corrected………thanks

L269: Please italicize “E. tenella”.

Response: corrected………thanks

Comments on the Quality of English Language

The English language is generally good, and the only issue is with the excessive use of "in this study" in the results section.

Response: corrected………thanks

Reviewer 2 Report

Comments and Suggestions for Authors

The manuscript contains important information regarding the effects of pommegranate peel extract to some parameters of intestinal health on broilers. On the other hand, I have doubts that the number of animals (n=300) is suitable to have further conclusions on performance traits (BW, FCR). In my opinion the number of pens per treatment is not enough for such estimstions, so I suggest to eliminate performance results from the manuscript. H&E  histological samples can be added instead. Please add the grant number of the experiment to the MM chapter, and the type of the applied normality test to the statistical analysis.

Comments on the Quality of English Language

English language is acceptable

Author Response

Reviewer 2

The manuscript contains important information regarding the effects of pommegranate peel extract to some parameters of intestinal health on broilers. On the other hand, I have doubts that the number of animals (n=300) is suitable to have further conclusions on performance traits (BW, FCR). In my opinion the number of pens per treatment is not enough for such estimstions, so I suggest to eliminate performance results from the manuscript. H&E histological samples can be added instead. Please add the grant number of the experiment to the MM chapter, and the type of the applied normality test to the statistical analysis.

Response: Dear academic reviewer, Thank you very much for your comments on our paper. The comments are highly constructive and have helped us improve our paper. We have revised the paper according to the valuable comments. The point by point is as follow: we have increased the number of birds number and replicates. Grant number has also been added at the end of the paper. Normality test has also been applied in the form of Kruskal–Wallis H test.

Reviewer 3 Report

Comments and Suggestions for Authors

Peer review report on ”Ameliorative Effect of Pomegranate Peel Powder on Growth Indices, Oocysts Shedding and Intestinal Health of Broilers Under Experimentally Induced Coccidiosis Condition” by Abdul Hafeez, Qambar Piral, Shabana Naz, Mikhlid H. Almutairi, Abdulwahed Fahad Alrefaei, Tugay Ayasan, Rifat Ullah Khan, Caterina Losacco.

General comments:

The authors examined the effect of ameliorative effect of pomegranate peel on growth traits, oocyst count, and intestinal histomorphology of broiler chickens and showed interesting results. However, I have a few suggestions.

Major comments:

1. How did the author choose the applied amount of pomegranate peel powder? Refer to the text, please.

2. The authors applied 3 and 6 g/kg amounts of pomegranate peel. In my opinion, this is a high amount. Did not it change the composition of the basal diet?

3. Would not it be more effective, if the authors applied a pure extract from the pomegranate peel?

4. It is confusing, that a simple summary talks about Japanese quails, but in the abstract (and other chapters) authors mention broiler chickens. Clarify in the text, please.

5. The authors do not mention how much Eimeria was used for inoculation either in the abstract or in the ’Birds husbandry and experimental design’ chapter.

6. In keywords, authors mention lemon peel powder. In my opinion, it is also very confusing.

7. In figure legends, Negative control always contains typos (Negitive control).

Comments on the Quality of English Language

Moderate editing of English language is required.

Author Response

Dear academic reviewer

Thank you very much for reviewing our paper along with the two other reviewers who also suggested minor suggestions. The comments you provided are highly constructive and have helped us to improve the quality of our paper. We have revised the paper in view of your comments. We hope that the revised paper will be acceptable to you. The point by point response is as follow:

Reviewer 3

Peer review report on ”Ameliorative Effect of Pomegranate Peel Powder on Growth Indices, Oocysts Shedding and Intestinal Health of Broilers Under Experimentally Induced Coccidiosis Condition” by Abdul Hafeez, Qambar Piral, Shabana Naz, Mikhlid H. Almutairi, Abdulwahed Fahad Alrefaei, Tugay Ayasan, Rifat Ullah Khan, Caterina Losacco.

General comments:

The authors examined the effect of ameliorative effect of pomegranate peel on growth traits, oocyst count, and intestinal histomorphology of broiler chickens and showed interesting results. However, I have a few suggestions.

Major comments:

  1. How did the author choose the applied amount of pomegranate peel powder? Refer to the text, please.

Response: The dose was chosen on the basis of search from the literature cited previously published…………thanks

  1. The authors applied 3 and 6 g/kg amounts of pomegranate peel. In my opinion, this is a high amount. Did not it change the composition of the basal diet?

Response: after addition of this amount of pomegranate peel, the feed was checked for analysis and there was no significant change in the composition of feed……..thanks

  1. Would not it be more effective, if the authors applied a pure extract from the pomegranate peel?

Response: Additional experiments are in progress, employing aqueous and methanolic extracts.

  1. It is confusing, that a simple summary talks about Japanese quails, but in the abstract (and other chapters) authors mention broiler chickens. Clarify in the text, please.

Response: corrected….thanks

  1. The authors do not mention how much Eimeria was used for inoculation either in the abstract or in the ’Birds husbandry and experimental design’ chapter.

Response: it is already there in the materials and methods, however, in the abstract it is added now…thanks

  1. In keywords, authors mention lemon peel powder. In my opinion, it is also very confusing.

Response: corrected…thanks

  1. In figure legends, Negative control always contains typos (Negitive control).

Response: corrected…..thanks

Moderate editing of English language is required.

Response: we have revised English of the paper throughout the paper.
